

# Comment on "Estimating the depth and evolution of intrusions at resurgent calderas: Los Humeros (Mexico)" by Urbani et al. (2020)

**Gianluca Norini** and **Gianluca Groppelli**

Istituto di Geologia Ambientale e Geoingegneria, Sezione di Milano, Consiglio Nazionale delle Ricerche, Italy

**Correspondence:** Gianluca Norini (gianluca.norini@cnr.it)

## Abstract

A multiple shallow–seated magmatic intrusions model has been proposed by Urbani et al. (2020) for the resurgence of the Los Potreros caldera floor, in the Los Humeros Volcanic Complex. This model predicts (1) the occurrence of localized bulges in the otherwise undeformed caldera floor, and (2) that the faults corresponding to different bulges exhibit different spatial and temporal evolution. Published data and a morphological analysis show that these two conditions are not met at Los Potreros caldera. A geothermal well (H4), located at the youngest supposed bulge (Loma Blanca) for which Urbani et al. (2020) calculated an intrusion depth (425±170 m), doesn't show any thermal and lithological evidence of such a shallow–seated cryptodome. Finally, published stratigraphic data and radiometric dating disprove the proposed common genesis of Holocene resurgence faulting and viscous lavas extruded in the centre of the caldera. Even if recent shallow intrusions may exist in the area, published data indicate that the pressurization of the LHVC magmatic/hydrothermal system driving resurgence faulting occurs at greater depth. Thus, we suggest that the model and calculation proposed by Urbani et al. (2020) are unlikely to have any relevance to the location, age and emplacement depth of magma intrusions driving resurgence at the Los Potreros caldera.

## 1 Introduction

Urbani et al. (2020) (henceforth U2020) made a contribution to the study of caldera resurgence based on field data and geothermal well logs from the Los Humeros Volcanic Complex (LHVC) and scaled analogue models. U2020 constrained the spatial–temporal evolution of post–caldera volcanism at LHVC and estimate the depth of the magmatic intrusions feeding the active geothermal system by integrating fieldwork data, well logs and laboratory results. The main conclusion of U2020 is that the resurgence of the Los Potreros caldera in the LHVC "*is due to multiple deformation sources*", "*linked to small magmatic intrusions located at relatively shallow depths (i.e. < 1 km)*". U2020 suggested that these intrusions are located below three uplifted areas surrounding the Arroyo Grande, Los Humeros and Loma Blanca faults, respectively.

The analysis by U2020 suffers from poor field data and contradictions with thermal remote sensing data (Section 2), geometric and structural inconsistencies between the LHVC post–caldera deformation and the analogue modelling (Section 3), lack of any substantial validation of the results with published well logs (Section 4), and incongruities with the reference stratigraphy and radiometric ages recently published by some of the U2020 authors (Section 5). These problems, which largely undermine the U2020 conclusions, are discussed below.



## 2 Location and relative age of faulting: field data and thermal remote sensing

U2020 analysed the occurrence and relative age of faulting, and proposed a new interpretation of some
structures identified by previous works, by studying faults and hydrothermal alteration in the Holocene
Cuicuiltic Member unit (Ferriz and Mahood, 1984; Arellano et al., 2003; Dávila–Harris and Carrasco–Núñez,
2014; Norini et al., 2015, 2019). The Cuicuiltic Member blankets the Los Potreros caldera floor (Fig. 1), is very
well exposed, has been dated at ca. 7 ka and is made of alternated fallout deposits of different composition
(Dávila–Harris and Carrasco–Núñez, 2014). The Cuicuiltic Member has been considered an ideal marker layer
for documenting Holocene faulting and stratigraphy in the caldera complex, because of the contrasting black
and white colours of the fallout deposits composing the unit (e.g. Ferriz and Mahood, 1984; Dávila–Harris
and Carrasco–Núñez, 2014; Norini et al., 2015, 2019; GEMex, 2019; U2020) (Figs. 1 and 2). The
reinterpretation by U2020 has been based on their field data (22 fault data in 3 outcrops), distinguishing
between lineaments ("*morphological linear scarps with no measurable fault offsets and/or alteration at the*
*outcrop scale*") and active and inactive faults ("*associated with measurable fault offsets and with active or*
*fossil alteration*"), respectively. The reinterpreted structures are the Las Papas, Las Viboras, Arroyo Grande
and Maxtaloya faults (Fig. 1).
We discuss the U2020 reinterpretation below, considering published field data (175 fault data in 24 outcrops,
Figs. 1 and 2, Tab. 1) and thermal remote sensing data (Fig. 3) (Norini et al., 2015, 2019; GEMex, 2019).

### *2.1 Las Papas and Las Viboras faults*

U2020 concluded that the Las Papas and Las Viboras are "*morphological scarps*" and "*lineaments*" not related
to faulting. For the Las Papas lineament, U2020 stated that "*unaltered and undeformed deposits of the*
*Cuicuiltic Member crop out along the E–W Las Papas lineament*" and that it "*is probably due to differential*
*erosion of the softer layers of the pyroclastic deposits*". Even if the Las Papas and Las Viboras structures were
several km long, the statements by U2020 have only been based on one outcrop on the Las Papas trace
(U2020 LH–08 outcrop, while the LH–07 outcrop is out of the fault trace; see Fig. 4C).
Several outcrops exist along the Las Papas and Las Viboras faults, as well as along many other faults in the
area surrounding these two main volcanotectonic structures (Fig. 1) (e.g. Dávila–Harris and Carrasco–Núñez,
2014; Norini et al., 2015, 2019; GEMex, 2019). In all these outcrops, the faults invariably displace the
Holocene Cuicuiltic Member and the underlying lava and pyroclastic units (Figs. 1 and 2; Tab. 1). These data
(Tab. 1) are incompatible with the U2020 conclusion that the Las Papas and Las Viboras are not faults. Indeed,
the data indicate that the Las Papas and Las Viboras structures have been originated in the Holocene by
faulting (Figs. 1 and 2, and Tab. 1) (Dávila–Harris and Carrasco–Núñez, 2014; Norini et al., 2015, 2019; GEMex,
2019). The U2020 description of their LH–08 outcrop can be explained by erosive retreat of the fault scarp, a
common process in dip–slip faults, especially in poorly consolidated sediments (e.g. Keller and Pinter, 2002;
Burbank and Anderson, 2011).

### *2.2 Arroyo Grande and Maxtaloya faults*

U2020 inferred that the Arroyo Grande and Maxtaloya scarps have been generated by nowadays inactive
faults. U2020 stated that these faults have been active "*prior to the deposition of the Cuicuiltic Member*". The
statement by U2020 arose from the analysis of two outcrops (their LH–09, see Fig. 4C, and the H6 well pad,
corresponding to the PDL08 outcrop of Figs. 1 and 2H), where "*strongly altered and faulted … lavas and*
*ignimbrites*" are "*covered by the unaltered Cuicuiltic Member*". Active/fossil alteration doesn't always allow
identifying faults or the age of faulting, because it depends also on their depth, life span of the hydrothermal





system, spatial relationships, and fluid paths along primary permeability and fracture zones (e.g. Bonali et al.,
2016; Giordano et al., 2016).
Outcrops of the Arroyo Grande and Maxtaloya faults show displacements of the Cuicuiltic Member, which
are incompatible with the conclusion of U2020 about the age of these two structures and the correlation
between faulting and hydrothermal alteration (Figs. 1 and 2; Tab. 1). The field data (Figs. 1 and 2, and Tab. 1)
indicate that the Arroyo Grande and Maxtaloya faults have been active after the deposition of the Cuicuiltic
Member (Dávila–Harris and Carrasco–Núñez, 2014; Norini et al., 2015, 2019; GEMex, 2019).
The Maxtaloya fault trace is coincident with a sharp thermal anomaly identified by Norini et al. (2015) (Fig.
3). U2020 didn't consider this positive (warm) anomaly when they discussed the thermal remote sensing
results published by Norini et al. (2015) (Section 5.3 in U2020). The thermal remote sensing data (Fig. 3)
suggest that the Maxtaloya fault plays nowadays an important role in the ascent of hot geothermal fluids
(Norini et al., 2015, 2019; Carrasco–Núñez et al., 2017; GEMex, 2019).
The Maxtaloya positive thermal anomaly constitutes the southern branch of a narrow warm corridor (T1 of
Norini et al., 2015), which is spatially coincident with the NNW–SSE fault swarm represented by the
Maxtaloya fault, Los Humeros fault and some sub–parallel normal and reverse fault strands (Fig. 3) (Norini et
al., 2019). This 7–8 km–long thermal anomaly is incompatible with the presence of the "*shallow and
delocalized heat sources*" proposed by U2020 (Fig. 3). Instead, the great length of this narrow thermal
anomaly is consistent with a deeper pressure source driving resurgence faulting (e.g. an asymmetric cup-
shaped intrusion), with lower surface temperatures in the centre of the thick resurgent block (cold area to
the east of the 7–8 km–long warm anomaly in Fig. 3) (see Norini et al., 2015).

## 3 Identification and geometry of uplifted areas: topographic data and structural mapping

U2020 identified three "*main uplifted areas*" surrounding the surface expressions of the Loma Blanca, Arroyo
Grande and Los Humeros faults. U2020 didn't provide any information on how these uplifted areas have
been identified and delimited with specific and reproducible criterion. The area around the Loma Blanca fault
has been named by U2020 "*Loma Blanca bulge*" and described as "*a morphological bulge, 1 km in width and
30 m in height*". The U2020 model also predicts the formation of an "*apical depression*" on top of a "*bulge*"
induced by a shallow magmatic intrusion. Indeed, U2020 depicted apical depressions on top of the three
"*uplifted areas*" of Loma Blanca, Arroyo Grande and Los Humeros (e.g. cross–sections in Fig. 10 by U2020).
Topographic profiles of the Los Potreros caldera floor extracted from a 1 m resolution Digital Elevation Model
(DEM) (Norini et al., 2019) show that the "*uplifted areas*" (or "*bulges*") identified by U2020 include
asymmetric reliefs and depressed sectors, and have boundaries not necessarily corresponding to slope
changes useful for their delimitation (Figs. 1 and 4A-C). The "*Loma Blanca bulge*" defined by U2020 comprises
a sector of a larger and uniform westward tilted and faulted surface (Norini et al., 2019). The western
boundary of the "*bulge*" is in the middle of the tilted surface, while the eastern one, corresponding to a
normal fault, is nearly at the same elevation of the summit of the "*bulge*" (Figs. 1 and 4A) (Carrasco–Núñez
et al., 2017; Norini et al., 2019). Similarly, the eastern and western boundaries of the Arroyo Grande and Los
Humeros "*uplifted areas*" have been located by U2020 in the middle of tilted or flat surfaces. The topographic
data extracted from the 1 m resolution DEM (Figs. 1 and 4A–B) are incompatible with the occurrence of the
"*main uplifted areas*" or "*bulges*" identified by U2020. The same topographic data are also incompatible with
the occurrence of any "*apical depression*" along the Arroyo Grande and Los Humeros faults, suggesting that





the present topography of the caldera floor doesn't have any relation with the "*uplifted areas*", "*bulges*" and
"*apical depressions*" presented by U2020 (Figs. 1 and 4A-C).
The analogue modelling by U2020 predicts the development of reverse faults at the base of the "*bulges*"
induced by the emplacement of shallow–seated cryptodomes (e.g. Fig. 7 by U2020). U2020 didn't provide
any field data or other evidence (morphostructural interpretation, geophysics, well logs, etc.) locating these
reverse faults, which are a fundamental feature of their model. Reverse faults of this type have been
identified in natural cases of shallow–seated intrusions (e.g. Sibbett, 1988; Jackson and Pollard, 1990;
Schofield et al. 2010; Wilson et al. 2016).
Structural maps of the Los Potreros caldera published by Carrasco–Núñez et al. (2017); Calcagno et al. (2018);
Norini et al. (2019) and U2020 are inconsistent with the idea of reverse faults at the base of the "*bulges*"
identified by U2020 (Figs. 1 and 4C). The "*Loma Blanca bulge*" is delimited to the east by a normal fault
mapped by Carrasco–Núñez et al. (2017) and Norini et al. (2019) (Fig. 4A).

**4 Validation of the proposed model: geothermal wells log data**
One of the most significant findings of U2020 is that the uplift in the "*Loma Blanca bulge*" has been generated
by a magmatic intrusion located at 425 ± 170 m of depth. U2020 also stated that this is the heat source of
the local geothermal anomaly. Such a shallow depth is within the range of geothermal wells drilled in the
area. A validation attempt of the U2020 model in the "*Loma Blanca bulge*" consists in the comparison of the
temperature and lithological H4 well log with the predicted intrusion depth. This well is located at the top of
the "*bulge*", just to the west of its "*apical depression*" (Fig. 4A,C). The H4 well log should show a significant
temperature change and intrusive/sub–volcanic lithologies at 425 ± 170 m of depth, if a shallow–seated, still
hot magmatic intrusion exists beneath the "*Loma Blanca bulge*".
According to data published by Arellano et al. (2003) and U2020, the H4 stratigraphic log doesn't show any
evidence of intrusive bodies from the surface down to 1900 m of depth, nor a sharp increase of the
temperature and geothermal gradient, which remains constant (about 20°C/100 m) (Fig. 4D). Also, the
temperature profiles measured in several wells of the field (e.g. Arellano et al., 2003) don't show any strong
temperature inversion or sharp change in the geothermal gradient possibly correlated to recent intrusive
bodies at very shallow depth ("*< 1 km*"), nor any shallow–seated intrusive/sub–volcanic lithology (Cavazos-
Álvarez et al., 2020). Lithological well logs show the presence of rhyolitic–andesitic rock layers within the
Caldera group (mainly in the Xaltipan ignimbrite unit; Carrasco–Núñez et al., 2017), which have been
interpreted by U2020 as "*intrusion of felsic cryptodomes within the volcanic sequence*". A recent study of
these felsic layers, based on petrographic and geochemical analyses of borehole samples, identified them as
"*lithic-rich breccias of local and irregular distribution that formed during the caldera collapse event*" (Cavazos-
Álvarez et al., 2020).
Published well log data indicate a deeper origin of the heat source (or sources) feeding the Los Humeros
geothermal field, with some variation of the temperature gradient due to faults and or permeability changes
(Fig. 4D) (e.g. Cedillo et al., 1997; Arellano et al., 2003; Cavazos-Álvarez et al., 2020).

**5 Validation of the proposed model: stratigraphic and radiometric data**
One of the results presented in U2020 is that "*…the recent (post–caldera collapse) uplift in the Los Potreros*
*caldera moved progressively northwards … along the Los Humeros and Loma Blanca scarps*". Based on the
proposed U2020 uplift model, it suggests that shallow intrusions of small magmatic bodies and,



consequently, the volcanic feeding system moved progressively northwards. This statement presents some
discrepancies with the stratigraphy, geological mapping and radiometric ages published recently (Carrasco–
Núñez et al., 2017, 2018; Juárez–Arriaga et al., 2018), as summarised by the following points:
a)   An obsidian dome (Qr1 Rhyolite of Carrasco–Núñez et al., 2017) has been dated using the U/Th
method at 44.8±1.7 ka by Carrasco–Núñez et al. (2017, 2018). Its location corresponds to the obsidian
dome cropping out along the Los Humeros fault described in U2020 and connected with the syn– to
post–Cuicuiltic Member eruption (7.3–3.8 ka) (Fig. 5). In U2020 there is no description of two
generations of obsidian domes along Los Humeros fault, nor any explanation to invalidate the
previous radiometric dating. Therefore, the U2020 attribution of this obsidian dome to the 7.3–3.8
176        ka volcanic activity phase appears unjustified and, consequently, weakens their model;

b)   The most recent volcanic activity of LHVC (post–Cuicuiltic Member) is clustered in two main ages,
around 3.8 and 2.8 ka, as indicated by recent radiometric and paleomagnetic data (Carrasco–Núñez
et al., 2017; Juárez–Arriaga et al., 2018) (Fig. 5). According to these ages and the LHVC geological map
(Carrasco–Núñez et al., 2017), the vents feeding the post–Cuicuiltic Member volcanic activity are
mainly located close to the southern and south–western sectors of the Los Humeros caldera rim.
These data suggest that the shallow feeding system of the post–Cuicuiltic Member activity is mainly
located in the southern and south–western sectors of the LHVC, some kilometres far from the
supposed bulged areas. Also, the ages and locations of the volcanic vents do not show any
progressive northward shift, but a scattered activity along the Los Humeros caldera rim.

**6 Conclusion**

We identified several problems in the U2020 study, showing that their model does not conform to most of
the published geological data about the Los Potreros caldera. The boundary conditions of a model and the
validation of the modelling results should always be based on the geological constraints that the natural
prototype imposes. In our opinion, the multiple magmatic intrusion model is imposed by U2020 to the natural
prototype regardless of several evidences of no fit between them. This mismatch between nature and model
includes the age and location of faulting, identification and delimitation of uplifted areas and apical
depressions, temperature and lithological wells log, and stratigraphic and radiometric data. The occurrence
of multiple magmatic intrusions at different depths in the Los Potreros caldera is not questioned in our
comment. Published data indicate that the calculations and conclusions by U2020 are unlikely to have any
relevance to the identification of the deformation source driving caldera resurgence and the heat source
feeding the geothermal field. The data and interpretations discussed in our comment have scientific and
economic implications. Indeed, they are important to plan the best strategies for geothermal exploration and
production, reducing drilling risk and potential loss of investment.

**Data availability:** all the data presented in this paper are available upon request.
**Author contributions:** Gianluca Norini and Gianluca Groppelli are equally responsible for *Conceptualization*
(research planning), *Investigation* (field analysis, geomorphological analysis, review of well logs and
stratigraphic data), *Writing - Original Draft* and *Review & Editing*, and *Visualization* (figures and maps).
**Competing interests:** the authors declare that they have no conflict of interest.



**Acknowledgments**
The research leading to these results has received funding from the GEMex Project, funded by the European
Union's Horizon 2020 research and innovation programme under grant agreement No. 727550. We thank
the Handling Topical Editor Joachim Gottsmann for providing valuable comments on an earlier version of the
manuscript. We also greatly thank Luca Ferrari and Joan Marti for the positive comments to our manuscript.

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

**Figure Captions**
**Figure 1:** volcanotectonic map of the Los Potreros caldera area, on a DEM (illuminated from the E) (modified
from GEMex, 2019 and Norini et al., 2019). Las V.F.: Las Viboras fault; Arroyo G.F.: Arroyo Grande fault; Loma
B.F.: Loma Blanca fault. Location of outcrops in Fig. 2 and Tab. 1 is shown. Traces of A–A'–A''–A''' and B–B'
topographic profiles of Fig. 4 are also shown.
**Figure 2:** photographs of faults in the Cuicuiltic Member along the structures mapped in Fig. 1.



**Figure 3:** enhanced surface kinetic temperature (SKT) of the Los Potreros caldera obtained from ASTER AST08
night–time thermal remote sensing data (see Norini et al., 2015, for details on methods and results).
Examples of field–validated sources of thermal anomaly are shown in the insets (Norini et al., 2015, 2019).
Thermal satellite data: credits LP-DAAC, USGS EROS data center at the NASA. Satellite images in the insets:
credits Esri, DigitalGlobe, GeoEye.
**Figure 4:** topographic profiles along the **(A)** A–A'–A''–A''' and **(B)** B–B' traces shown in Fig. 1, and **(C)** schematic
geological map (modified from U2020) outlining the three uplifted areas discussed by U2020; the traces of
the two topographic profiles and the locations of the H4 and H20 well are also shown. **(D)** H4 lithological and
temperature log (well data from Arellano et al., 2003, and U2020). P.c.: Post–caldera group.
**Figure 5:** map of the post–Cuicuiltic Member vents and ages based on radiometric data, paleomagnetic
analysis or inferred from geological map (Carrasco–Núñez et al., 2017, 2018; Juárez–Arriaga et al., 2018). The
post–Cuicuiltic Member uplifted areas and obsidian dome proposed by U2020 are also shown. Active faults
are from Norini et al. (2019).
**Table 1** (supplementary data): field data of faults and fractures deforming the Cuicuiltic Member and the
underlying units (see Fig. 1 for outcrops location).




Figure 1:



Figure 1 map showing elevation and resurgence structures.





Figure 2:

![A](LH1732 (19° 39' 50.981" N, 97° 25' 44.558" W))

![B](LH64 (19° 40' 25.108" N, 97° 25' 45.633" W))

![C](LH62 (19° 40' 18.786" N, 97° 26' 6.535" W))

![D](LH66 (19° 40' 38.931" N, 97° 26' 0.571" W))

![E](LH23 (19° 40' 28.558" N, 97° 26' 38.304" W))

![F](PDL52 (19° 40' 41.187" N, 97° 26' 35.606" W))

![G](LH20 (19° 39' 15.969" N, 97° 26' 31.245" W))

![H](PDL08 (19° 39' 2.290" N, 97° 26' 25.369" W))







Figure 3:

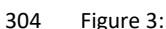





Figure 4:

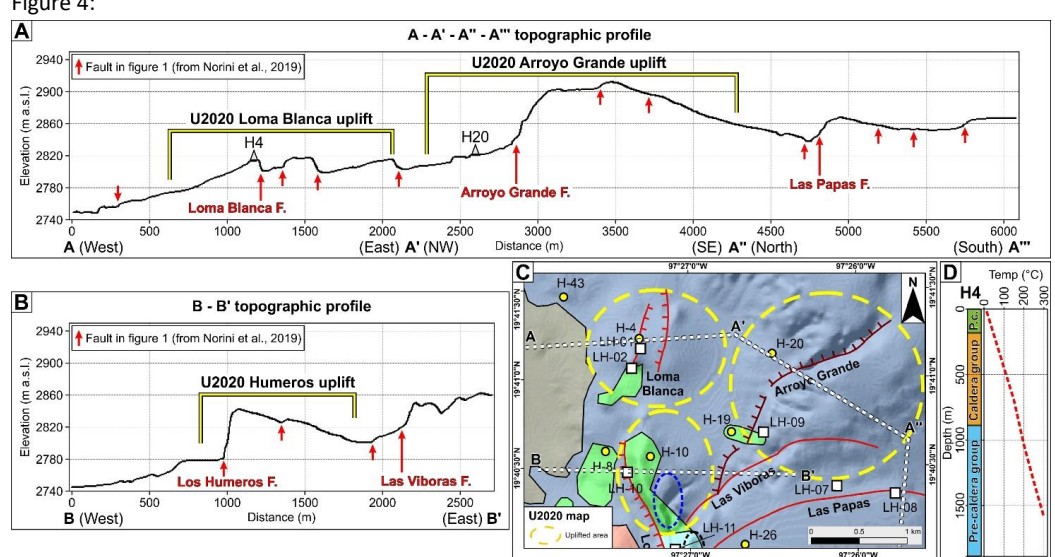



Figure 5:

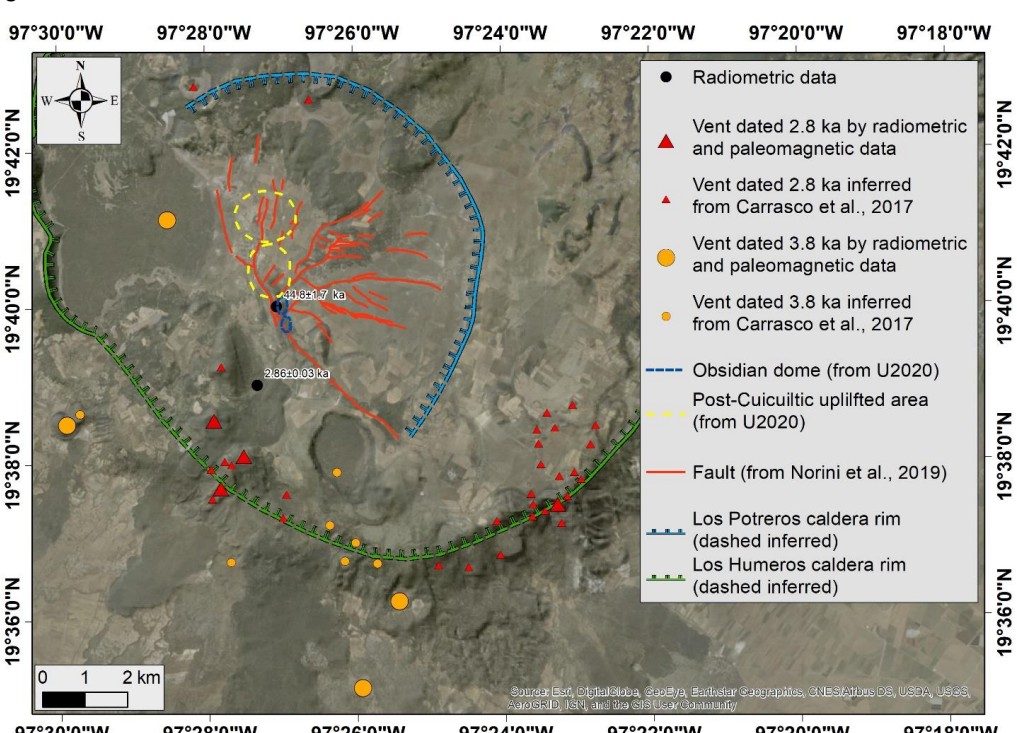

