# Peer review of "Comment on "Estimating the depth and evolution of intrusions at resurgent calderas: Los Humeros (Mexico)" by Urbani et al. (2020)"

_Solid Earth, 2020_

## Referee Comment (RC1) · Joan Marti (Referee) · 10 Oct 2020

As I indicated in my previous review of the comment made by Norini and Groppelli on the paper of Urbani et al (2020), this is well written and documented comment that challenges some of the interpretations and conclusions of the original paper. This new version of the comment is more polite and less aggressive than the previous one, and states on those points of disagreement with the Urbani et al (2020) paper. Repeating what I said in my former review, Norini and Groppelli show a good field knowledge of the study area and on the previous literature, and use it to discuss some of the results and interpretations presented by Urbani et al (2020). The main contradictions

are found concerning the interpretation of some faults, the identification and geometry of uplifter areas inside the caldera, the validation of the proposed model of Urbani et al (2020) with well logs, and the stratigraphy and radiometric ages they present. The arguments presented by Norini and Groppelli generate doubts about the work done by Urbani et al (2020), at least for what concerns their interpretation of the data presented and, particularly, the conceptual model proposed. I fully agree with Norini and Groppelli when they say that "the boundary conditions of a model and the validation of the modelling results should always be based on the geological constraints that the natural prototype imposes", so if there is doubt on the suitability and accuracy of the geological constraints used, the resulting model may not be reliable and, therefore, needs revision. I feel that this revised version of the Norini and Groppelli's comment should be published as it is.

―――――――――――――――――――――――

---

## Short Comment (SC1) · 13 Oct 2020

As I expressed in my previous review, I consider that Norini and Groppelli provided solid geological information that question the model presented in Urbani et al. (2020), although the first version at places could have been a bit overtone. In this new version the authors smoothed the parts that may have been considered unappropriated for a scientific debate and provide more details that confirm their in-depth knowledge of the field geology and tectonics of the Los Humeros caldera. Their comment is well written and illustrated. It is also properly organized with precise explanation on four issues that question the model of Urbani et al. The first and main argument is about location and

relative age of faulting and include new material. Here Norini and Groppelli provide a wealth of field data, now also in table format, and a new, remote sensing-based thermal anomaly map, which does not support the Urbani et al. model. The other three sections are essentially the same as the previous version and show how topographic data, geothermal well log data, and volcanic stratigraphy supported by isotopic ages are also at odds with the Urbani et al. model. I think that the comment has solid scientific bases and the questioning is now expressed in a polite form so that it can be accepted for publication as is.

―――――――――――――――――――――

---

## Editor Comment (EC1) · Joachim Gottsmann (Editor) · 27 Oct 2020

The paper in its original version has been seen and evaluated by two reviewers who independently recommended publication of the comment on scientific grounds. As both Executive and Handling Editor I asked for the comment paper to be resubmitted after toning down some of the wording. This has now happened and I have no additional criticisms that would prevent the publication of this comment. The authors of "Estimating the depth and evolution of intrusions at resurgent calderas: Los Humeros (Mexico)" by Urbani et al (2020) will now have the opportunity to respond to the comment.

---

## Author Comment (AC1) · 17 Nov 2020

We greatly thank Joan Marti for the positive comments to our manuscript. Best Regards, Gianluca Norini

---

## Author Comment (AC2) · 17 Nov 2020

We greatly thank Luca Ferrari for the positive comments to our manuscript. Best Regards, Gianluca Norini

---

## Author Comment (AC3) · 17 Nov 2020

We greatly thank Joachim Gottsmann for the positive comments to our manuscript. Best Regards, Gianluca Norini
* * *

---

## Author Response (AR1)

**Ref: SE-2020-168**
Author's response

We would like to thank the Reviewers and the Handling Editor for the positive comments to our manuscript.

Following their suggestions, we have decided to do not make changes to our manuscript.

I sign this letter on the behalf of all the Authors of the manuscript ID **SE-2020-168**.

Best regards,

Gianluca Norini
Istituto di Geologia Ambientale e Geoingegneria
Consiglio Nazionale delle Ricerche
Area della Ricerca CNR - ARM3
Via Roberto Cozzi 53
20125 Milano
Italia